# Correlations between Convenience Cooking Product Use and Vegetable Intake

**DOI:** 10.3390/nu14040848

**Published:** 2022-02-17

**Authors:** Natasha Brasington, Tamara Bucher, Emma L. Beckett

**Affiliations:** 1School of Environmental and Life Sciences, The University of Newcastle, Ourimbah, NSW 2258, Australia; natasha.brasington@uon.edu.au (N.B.); tamara.bucher@newcastle.edu.au (T.B.); 2Priority Research Centre for Physical Activity and Nutrition, The University of Newcastle, Callaghan, NSW 2308, Australia; 3Hunter Medical Research Institute, New Lambton Heights, NSW 2305, Australia

**Keywords:** meal bases, recipe bases, convenience cooking products, vegetables, recipes, cooking sauces

## Abstract

Australians’ vegetable intakes are low, and strategies are needed for improvement. Popular convenience cooking products (meal bases and recipe bases, ready-made marinades, and convenience cooking sauces) address common cooking and vegetable consumption barriers (cost, time, and cooking skills). However, relationships between their usage and vegetable intakes have not been established. Therefore, Australian adults were surveyed on convenience cooking product use, vegetable intake and variety, behaviours when barriers to vegetable inclusion arise, and vegetable choice factors. Of 842 participants, 36.7% used meal and recipe bases, 28.1% marinades, and 47.2% cooking sauces, with most following the back-of-pack recipes at least sometimes. A total of 12.5% of participants used products from all three categories. Factors associated with lower vegetable intakes were meal and recipe base and cooking sauce use, using a higher number of product categories, and always following back-of-pack recipes. Factors associated with lower vegetable variety were the use of meal and recipe bases and cooking sauces. Factors in vegetable choice, and behaviours when not including a listed vegetable (due to not having or liking the vegetable, or an inability to eat it) did not vary by usage habits. These results provide insights into current vegetable intakes of those using convenience products, providing a baseline for future changes in the product design and recommendations.

## 1. Introduction

It is well established that diets regularly containing a high level and variety of vegetables are beneficial to overall health, reducing risks for diet-related chronic diseases [1,2,3]. As such, the World Health Organization recommends consuming 400 g of vegetables per day [4]; similarly, the Australian Guide to Healthy Eating suggests that most adults eat a minimum of 375 g of vegetables a day (at least five serves of vegetables at ~75 g per serve) [5]. Recommendations also specify that intake should include a variety of vegetables, as each vegetable has a unique profile of nutrients and other healthful compounds [6]. Nevertheless, the results of the 2018 Australian National Health Survey demonstrated that ~95% of participants did not meet these recommendations [7]. Importantly, despite multiple public education campaigns, this figure had not changed over the ten preceding years [8].

There is an array of barriers that can contribute to low and limited vegetable intake [9]. These include a lack of availability, time, and access, as well as burdens of cost and taste preference [10,11,12,13]. It has been noted that, in the broader population, cooking skills have been at a decline, and devalued, which is a concern for individual food choices and health outcomes [14]. Importantly, vegetables often require preparation or cooking prior to consumption, and so cooking skills and confidence may present an additional barrier for vegetables compared to other healthy food groups such as fruits [15,16]. As such, low levels of cooking skills can leave individuals vulnerable to messages of how the products can be easier to prepare, and how they may taste [11], rather than messages regarding the nutritional quality or vegetable contents of these products. Convenience foods and ready-to-eat meals prepared out of the home are typically calorie-dense compared to home cooking, which is associated with healthier dietary choices, and higher intake of fruits, vegetables, and whole grains [17,18].

However, convenience products were developed to increase convenience while encouraging home cooking. The term “convenience cooking products” refers to products that provide a base for cooking, including recipes with suggested ingredients, which lower meal preparation and cooking times. Examples include meal and recipe bases; ready-made marinades; and convenience cooking sauces, including pasta sauces and simmer sauces. These products have become mainstream in Australian households in recent times [19]. It has also been suggested that the popularity of convenience products is not only to do with time, but also in reducing the physical and cognitive effort regarding planning and preparing meals [20,21]. Those in Australia who report using convenience cooking products also scored lower on scales of cooking confidence and creativity [22].

The nutritional quality of convenience cooking products is often perceived as being non-nutritious, energy-dense, high in fats and carbohydrates, and lacking essential micronutrients [23,24,25]. However, these products do have back-of-pack recipes provided on them listing vegetables, protein, and/or grains which are to be added by the consumer to create the final meal [26]. Therefore, if these recipes are followed, they may be a tool used to surmount barriers to vegetable consumption, subsequently improving diet quality.

However, there is no independent data on the relationship between the use of any category of convenience cooking products, the following of the recipes provided, and vegetable consumption. Therefore, we conducted a cross-sectional survey study using scales of factors determining vegetable choices, and collected information on convenience cooking product use (meal and recipe bases, ready-made marinades, and convenience cooking sauces), frequency of following the recipes provided on the products, consumers’ vegetable consumption and their determining factors for vegetable consumption.

## 2. Materials and Methods

### 2.1. Study Desgin and Recruitment

This cross-sectional survey (via Qualtrics, SAP, Provo, UT, USA) was conducted using snowball recruitment, as described in Brasington et al., for ~7 weeks in 2020 [22]. Participants were living in Australia, over 18 years of age, and proficient in English comprehension skills, as the survey was administered in English. Ethics approval was granted by the Human Research Ethics Committee of the University of Newcastle (Reference No. H-2020-0119). The online questionnaire used both qualitative and quantitative questions developed by the authors, and was piloted internally. Questions were arranged in thematic blocks based on consumption of vegetables, factors determining vegetable choices, convenience cooking products usage habits, and demographics.

The participants were asked to self-identify if they used convenience cooking products (“Do you use any of the following products?”) with several product types listed (to ensure recognition across different brands) and collapsed into the following categories for analysis: meal and recipe bases (including meal bases, recipe bases, and recipe concentrates), ready-made marinades (marinades only), and convenience cooking sauces (including simmer sauces, pasta sauces, and other sauces), or none of the above. The number of product categories used was also summed. Participants were then categorised as “users” or “non-users” of these products. Regarding back-of-pack recipes, participants were asked if they always, sometimes, or never follow the recipes provided with products [22]. In a matrix style question, users were also asked to select from a list of actions what they were most likely to do if they were following a back-of-pack recipe on the product classes they reported using and either did not have, did not like, or could not eat a vegetable listed (not including that vegetable, replacing it with a different vegetable, replacing it with a similar vegetable, not making the meal at all, not sure, other).

Participants were asked to report their typical daily vegetable consumption in serves per day (with a serves guide provided) using the question “How many serves of vegetables do you typically eat per day (1 serve of vegetables is half a cup of cooked vegetables or 1 cup of salad)?”, and a numerical dropdown provided. Responses were accordingly categorised into either meeting or not meeting the daily recommendation for vegetables as per the Australian Guide to Healthy Eating. As an indicator of vegetable variety and consumption, eating frequency of fourteen of the most commonly consumed vegetables in Australia was assessed, based on the Australian Healthy Eating Quiz [27]. Participants were then given a vegetable variety score (score 1 for each vegetable they consumed at least one serve of regularly, i.e., at least weekly), with a possible range of scores from 0–14.

Factors involved in determining vegetable selection were rated on a 5-point Likert scale (ranging from 1 = not at all important to 5 = extremely important). There were 12 factors rated (taste, flavour, costs, sustainability, availability, texture, colour, nutritional content, easy to cook, shelf life, quality [28], and value for money).

### 2.2. Statistical Analysis

Data were analysed using the statistical analysis software package JMP (Pro 14; SAS Institute Inc., Cary, NC, USA). The threshold for statistical significance was set at a *p*-value of <0.05, and *p*-values were reported to one significant figure. Contingency tables (Pearson χ^2^) and nominal logistical regression were used to assess the differences in distributions between categories. T-tests and ANOVA with Tukey’s post-hoc tests were used to compare the differences in continuous variables between groups. It was hypothesised that those who used convenience cooking products would have lower vegetable intakes and variety scores than those who never used them, but that the following of the recipes would be linked to higher vegetable intakes and variety scores.

## 3. Results

A total of 964 people gave informed consent, and participated in the survey. One hundred and twenty-two of those participants were excluded due to incomplete responses, or for having completed the survey in less than half the median completion time (244 s), or for failing the attention check (similar questions with reversed scales). Overall, there was a total of 842 responses included in the final sample.

The age of respondents ranged from 18–80 years (median 41 years, standard deviation 12.1 years). The sample was mostly female (77.7%), and the majority had a university level of education (Table 1). The majority had household incomes above AU$75,000 per year (equivalent to approximately €47000), and reported working full-time hours (30 h per week or more; Table 1). 36.7% of participants reported using meal and recipe bases, 28.1% used marinades, and 47.2% used cooking sauces (including pasta sauces and simmer sauces). There were no differences in incomes, education, work hours, or sex distributions between the user and non-user groups for any of the convenience cooking products. Those who used convenience cooking sauces cooked at home less frequently than non-users (*p* = 0.003, Table 1). Regarding the number of individual convenience cooking product classes used, 41% of participants used none, 23.5% used only one product type, 22.5% used two product types, and 12.6% used all three.

### 3.1. Following of Back-of-Pack Recipes

The majority of participants who used meal and recipe bases, marinades, and cooking sauces followed the recipes provided on the back of the pack at least sometimes (sometimes (41.9–44.1%) or always (21.9–24.9%)); however, 32.7–34.2% of participants never followed the recipes provided (Table 2).

### 3.2. Reported Vegetable Intake and Variety Scores

Mean vegetable intake in the sample was 3.1 serves per day (standard deviation = 1.2 serves per day). Users of meal and recipe bases were less likely to meet the daily recommended vegetable intakes than non-users (10.4% vs. 17.3%, respectively: χ^2^ = 7.4, *p* = 0.006). Similar results were found for users of convenience cooking sauces (10.8% vs. 18.2%; χ^2^ = 9.0, *p* = 0.003). However, there were no differences in the proportions meeting the recommended thresholds in ready-made marinade users compared to non-users. When reported daily vegetable serves were considered as a continuous variable, those who reported using any of the convenience cooking products (meal and recipe bases, marinades, or cooking sauces) had lower reported mean daily vegetable intakes compared to non-users (*p* ≤ 0.005, Table 3). However, there was no significant difference between vegetable variety scores between users and non-users of any product class (Table 3).

Among meal and recipe base users, those who reported never following the recipes were the most likely to meet the daily recommended intakes (16.0%) compared to those who reported always (5.9%) and sometimes (6.9%; χ^2^ = 8.0, *p* = 0.002) following the recipes. Similar results were found for users of convenience cooking sauces (never = 15.4%, always = 4.35%, and sometimes = 9.2% meeting recommended intakes; χ^2^ = 8.8, *p* = 0.01). However, there were no differences in the proportions meeting the recommended thresholds by recipe-following for marinade users (*p* = 0.1). When reported daily vegetable serves were considered as a continuous variable, meal and recipe base users who reported always following the recipes had lower reported mean daily vegetable intakes compared to sometimes and always followers (*p* = 0.0001, Table 4). Users of meal and recipe bases who reported always following the recipes also had lower vegetable variety scores than those who never or sometimes followed (*p* = 0.0001; Table 4). For convenience cooking sauces, those who always followed the recipes had lower vegetable intakes and variety scores than those who sometimes followed (*p* = 0.0008 and *p* = 0.009, respectively; Table 4). However, intakes and variety scores did not differ by recipe-following for marinade users (*p* = 0.7; Table 4).

Those who did not report using any of the convenience cooking products had higher vegetable intakes than those who used two or three product classes (Table 5). However, vegetable variety score did not vary by number of product categories used (Table 5).

Those who did not report using any of the convenience cooking products were the most likely to meet the daily recommended intakes of vegetables (19.5%) compared to those using one product class (11.1%), two product classes (13.8%), or three product classes (7.6%; χ^2^ = 12.8, *p* = 0.005).

### 3.3. Decisions When Barriers Arise to Including the Vegetables Specified on Back-of-Pack Recipes

Users of convenience cooking products were asked what they would do if they did not have, could not eat, or did not like vegetables listed in the recipes provided with meal and recipe bases to assess their behaviours when faced with these barriers. The vast majority of participants reported that they would replace that vegetable with another vegetable or a similar one if they did not have (87.4–88.7%), did not like (85.2–89.0%), or could not consume (81.9–87.7%) the vegetable listed (Table 6). Not including the vegetable and not making the meal at all were the least common courses of action. These results were similar when analysed by number of convenience cooking product categories used (Table 7).

### 3.4. Factors Determining Vegetable Choices

The top three determinants of choice regardless of convenience cooking product use or number of categories of product used were taste, flavour, and quality (Table 8 and Table 9). The three least important determinants were shelf-life, sustainability, and colour (Table 8 and Table 9). There were no significant differences in the mean levels of importance given to each factor in determining vegetable choices between those who reported using each convenience cooking product and those who did not, other than a small difference in the importance score for shelf-life in convenience cooking sauce users compared to non-users (Table 8), and sustainability between the number of convenience cooking products used (Table 9).

## 4. Discussion

This study is the first to investigate the relationships between usage habit key examples of convenience cooking products, and vegetable consumption habits. It appears that the use of these products, both in binary terms (user relative to non-users) and in terms of using a higher number of convenience cooking product categories, was associated with lower vegetable intakes. This may reflect the fact that those who choose convenience cooking products face more barriers to obtaining a healthy diet and accessing vegetables. We have previously demonstrated that those who use convenience cooking products (including meal and recipe bases, marinades, and other sauces) have lower cooking confidence and creativity [22], and it is established that those with lower cooking skills are less likely to eat a healthy balanced diet [29,30,31]. Additional research is needed to determine if convenience cooking products are acting as a tool to replace other less healthful convenience foods, such as takeaway foods and ready-made meals accessed outside of the home, or if they are displacing more healthful home-cooked meals, in order to assess the utility of these products in encouraging a healthy diet.

Furthermore, those who used meal and recipe bases and other cooking sauces, and always followed the recipes provided on the back of the pack had lower vegetable intakes and variety scores than those who never followed the recipes. Again, this may be linked to cooking confidence and creativity, with those with low creativity and confidence more likely to follow recipes. However, we have also previously conducted an audit of the back-of-pack recipes provided with meal and recipe bases available at major Australian supermarket chains, and demonstrated that these products are typically low in total serves of vegetables, and are low in vegetable variety, featuring mostly starchy and orange vegetables, and low levels of green vegetables [26]. Therefore, reformulation of these back-of-pack recipes to include more vegetables, or to recommend side salads, could help to improve vegetable intake in those who use these products and follow the recipes. These associations were not found for users of ready-made marinades, and as such, more investigation may be needed to distinguish between market segments of different convenience cooking products.

Importantly, though a total of 76.2–78.1% of participants reported only following the recipes sometimes or never, the vast majority of participants reported that when they did not have, could not eat, or did not like a vegetable ingredient listed, they would replace that ingredient with another vegetable (either a different vegetable or similar). This means that manufacturers can be confident that consumers are obtaining the serves of vegetables listed in the recipes when designing these recipes to encourage vegetable intakes. The deviation to different vegetables may also explain why those who do not always follow the recipes have higher vegetable variety scores.

Though time, knowledge, and cost are commonly cited barriers to vegetable consumption [9,10,11,12,13,14,15,16], income, education, work hours, and selection factor variables did not vary by usage or recipe-following habits. However, it is important to note that this sample was collected as a snowball sample and convenience cohort, resulting in a selection bias toward women with high incomes and education levels, as is typical for this method [18]. As such, these findings require validation in a more balanced sample. However, women remain disproportionately responsible for home cooking and grocery shopping in Australia [32,33], and so this sample may be biased toward representing those who are actively involved in cooking and grocery decision-making in Australia.

Other limitations include the self-reporting of vegetable intake, which is vulnerable to over-reporting [34], without prompts as to what to consider as a vegetable, which may conversely result in under-reporting. Classification as users and non-users based on self-reporting also lacks resolution, as frequency of use was not captured, and this may impact results. This requires further investigation. Frequency of recipe-following also requires more resolution in futher investigation to quantify potential differences in “sometimes” followers. However, strengths include the large sample size, the multiple measures of vegetable consumption (intake and variety), the multiple categories of convenience cooking product investigated, and the novelty of the questions, with potential implications for consumers, industry, and public health. The findings presented here justify further research into the role convenience cooking products play in diet quality.

Though convenience cooking products have the potential to help increase vegetable consumption in users, additional research is needed to provide a deeper insight into consumers’ attitudes and actions towards vegetable consumption when using these products. A better understanding of usage habits and vegetable consumption drivers may assist the industry in adapting these products accordingly, to ensure an adequate vegetable intake and variety of vegetable intake. Overall, traditional methods of education and promotion have not improved population vegetable intakes at a population scale; as such, shifting paradigms are still needed to improve vegetable intakes, using products that meet consumer needs, and address common barriers to home cooking. The presented data, showing lower vegetable intakes with product use, with a higher number of product types used, and with more frequent recipe-following, but with little difference in related vegetable choice factors, provide an insight into the convenience cooking product users’ vegetable intake, which provides a baseline for future improvements to the product back-of-pack recipe vegetable content in hopes to see an increase in the users’ vegetable consumption.

## Figures and Tables

**Table 1 nutrients-14-00848-t001:** Demographics of the total sample, and by the use of convenience cooking products of interest.

	Total	Meal and Recipe Bases			Marinades			Cooking Sauces		
		Users	Non- Users	χ^2^ (*p*)	Users	Non- Users	χ^2^ (*p*)	Users	Non- Users	χ^2^ (*p*)
	n (%)				n (%)			n (%)		
Sex										
Male	171 (20.3)	75 (24.3)	96 (18.0)	4.7 (0.09)	53 (22.4)	118 (19.5)	0.8 (0.6)	82 (21.1)	89 (20.0)	2.2 (0.3)
Female	654 (77.7)	228 (73.8)	426 (79.9)		179 (75.5)	475 (78.5)		304 (76.6)	350 (78.7)	
Others	17 (2.0)	6 (1.9)	11 (2.1)		5 (2.1)	12 (2.0)		11 (2.3)	6 (1.3)	
Income										
<$20,000	25 (3.0)	6 (1.9)	19 (3.6)	8.2 (0.4)	4 (1.7)	21 (3.5)	2.8 (0.7)	12 (3.0)	13 (2.9)	0.3 (0.9)
$20,000–$49,999	88 (10.5)	38 (12.3)	50 (9.4)		28 (11.8)	60 (9.9)		42 (10.5)	46 (10.3)	
$50,000–$74,999	84 (10.0)	27 (8.7)	57 (10.7)		25 (10.5)	59 (9.8)		41 (10.3)	43 (9.7)	
$75,000–$149,999	303 (36.0)	122 (39.5)	181 (34.0)		84 (35.4)	219 (36.2)		141 (35.5)	162 (36.4)	
>$150,000	235 (27.9)	84 (27.2)	151 (28.3)		67 (28.2)	168 (27.8)		112 (28.2)	123 (27.6)	
Others *	107 (12.7)	32 (10.4)	75 (14.1)		29 (12.2)	78 (12.9)		49 (12.3)	58 (13.0)	
Working hours/week										
<15	173 (20.5)	60 (19.7)	113 (21.3)	1.3 (0.7)	48 (20.4)	125 (20.8)	2.2 (0.5)	80 (20.4)	93 (21.0)	0.09 (0.9)
15–30	151 (17.9)	54 (17.7)	97 (18.3)		38 (16.1)	113 (18.8)		71 (18.1)	80 (18.1)	
30–50	455 (54.0)	167 (54.7)	288 (54.3)		129 (54.9)	326 (54.3)		216 (55.0)	239 (54.1)	
50+	56 (6.7)	24 (7.9)	32 (6.0)		20 (8.5)	36 (6.0)		26 (6.6)	30 (6.8)	
Education										
<Year 12	28 (3.3)	13 (4.2)	15 (2.8)	5.7 (0.3)	11 (4.6)	17 (2.8)	8.5 (1.3)	12 (3.0)	15 (3.6)	8.2 (0.1)
Year 12	89 (10.6)	37 (12.0)	52 (9.8)		31 (13.1)	58 (9.6)		43 (10.8)	46 (10.3)	
Technical diploma	116 (13.8)	46 (14.9)	70 (13.1)		31 (13.1)	85 (14.1)		57 (14.4)	59 (13.3)	
Degree	290 (34.4)	107 (34.6)	183 (34.3)		85 (35.9)	205 (33.9)		137 (34.5)	153 (34.4)	
Postgrad.	314 (37.3)	103 (33.3)	211 (39.6)		76 (32.1)	238 (39.3)		143 (36.0)	171 (38.4)	
Nights cooking at home/week										
>7	250 (29.7)	81 (26.2)	169 (31.7)	4.9 (0.3)	59 (24.9)	191 (31.6)	3.8 (0.4)	96 (24.2)	154 (34.6)	15.6 (0.003)
5–6	310 (36.8)	114 (36.9)	196 (36.8)		92 (38.8)	218 (36.0)		149 (37.5)	161 (36.2)	
3–4	201 (23.9)	85 (27.5)	116 (21.8)		62 (26.2)	139 (23.0)		102 (25.7)	99 (22.2)	
1–2	68 (8.1)	25 (8.1)	43 (8.1)		20 (8.4)	48 (7.9)		42 (10.6)	26 (5.8)	
<1	13 (1.5)	4 (1.3)	9 (1.7)		4 (1.7)	9 (1.5)		8 (2.0)	5 (1.1)	

* Others = did not know, or declined to respond.

**Table 2 nutrients-14-00848-t002:** Reported frequency of back-of-pack recipe-following for convenience cooking products of interest in users of convenience cooking products.

	Always	Sometimes	Never
	%		
Meal/recipe bases	24.9	42.4	32.7
Marinades	21.9	43.9	34.2
Cooking sauces	23.2	44.1	32.7

**Table 3 nutrients-14-00848-t003:** Daily vegetable intakes and vegetable variety scores by use of convenience cooking products of interest.

	Vegetable Serves/Day			Vegetable Variety Score		
	Users	Non-Users	*p*	Users	Non-Users	*p*
	Mean (95% CI)			Mean (95% CI)		
Meal/recipe bases	2.8 (2.6–2.9)	3.2 (3.1–3.3)	<0.0001	8.6 (8.4–8.9)	8.9 (8.7–9.1)	0.1
Marinades	2.9 (2.7–3.0)	3.1 (3.0–3.2)	0.005	9.0 (8.7–9.3)	8.8 (8.6–9)	0.7
Cooking sauces	2.9 (2.7–3.0)	3.2 (3.103.3)	<0.0001	8.7 (8.4–8.9)	9.0 (8.7–9.2)	0.08

CI = confidence interval.

**Table 4 nutrients-14-00848-t004:** Daily vegetable intakes and vegetable variety frequency of back-of-pack recipe-following in users of convenience cooking products of interest.

	Vegetable Serves/Day				Vegetable Variety Score			
	Always	Sometimes	Never	F (*p*)	Always	Sometimes	Never	F (*p*)
	Mean (95% CI)				Mean (95% CI)			
Meal/recipe bases	2.4 ^a^ (2.1–2.6)	2.7 ^b^ (2.5–3.0)	3.1 ^c^ (2.9–3.3)	9.3 (0.0001)	7.5 ^a^ (6.8–9.2)	8.7 ^b^ (8.2–9.2)	9.3 ^b^ (8.8–9.7)	8.3 (0.0001)
Marinades	2.7 ^a^ (2.4–3.0)	3.0 ^a^ (2.8–3.2)	2.8 ^a^ (2.5–3.1)	1.2 (0.7)	8.7 ^a^ (8.0–9.4)	9.3 ^a^ (8.8–9.8)	8.8 ^a^ (8.2–9.4)	1.2 (0.6)
Cooking sauces	2.5 ^a^ (2.2–2.7)	3.1 ^b^ (2.9–3.3)	2.9 ^a,b^ (2.7–3.1)	7.2 (0.0008)	7.9 ^a^ (7.4–8.5)	8.9 ^b^ (8.6–9.3)	8.8 ^b^ (8.4–9.3)	4.8 (0.009)

Means in the same group not connected by the same letter are statistically different (*p* < 0.05). CI = confidence interval.

**Table 5 nutrients-14-00848-t005:** Daily vegetable intakes and vegetable variety by number of convenience cooking product categories used.

	Vegetable Serves/Day		Vegetable Variety Score	
	Mean (95% CI)	*p*	Mean (95% CI)	*p*
Use no product class	3.27 (3.14–3.40) ^a^	<0.0001	8.99 (8.73–9.25) ^b^	0.5
Use one product class	3.03 (2.86–3.19) ^a,b^		8.70 (8.35–9.05) ^b^	
Use two product classes	2.88 (2.69–3.07) ^b^		8.69 (8.30–9.08) ^b^	
Use three product classes	2.69 (2.47–2.91) ^b^		8.79 (8.28–9.31) ^b^	

Means in the same group not connected by the same letter are statistically different (*p* < 0.05). CI = confidence interval.

**Table 6 nutrients-14-00848-t006:** Actions taken when barriers to including a vegetable listed on a convenience cooking product back-of-pack recipe arises, by convenience cooking product usage.

	Meal and Recipe Bases			Marinades			Cooking Sauces		
	Do Not Have	Do Not Like	Cannot Consume	Do Not Have	Do Not Like	Cannot Consume	Do Not Have	Do Not Like	Cannot Consume
	n (%)								
Not include that vegetable	26 (8.4)	30 (9.7)	35 (11.3)	19 (8.0)	17 (7.2)	20 (8.4)	31 (1.8)	37 (9.3)	44 (11.1)
Replace (different vegetable)	140 (45.3)	176 (57.0)	169 (54.7)	112 (47.3)	152 (64.1)	148 (62.4)	197 (49.6)	246 (62.0)	233 (58.7)
Replace (similar vegetable)	130 (42.1)	87 (28.2)	84 (27.2)	98 (41.4)	59 (24.9)	60 (25.3)	152 (38.2)	97 (24.4)	94 (23.7)
Not make the meal	9 (2.9)	11 (3.6)	13 (4.2)	5 (2.1)	6 (2.5)	7 (2.9)	12 (3.0)	12 (3.0)	17 (4.3)
Not sure	2 (0.7)	1 (0.3)	4 (1.3)	2 (0.8)	1 (0.4)	1 (0.4)	1 (0.2)	0 (0.0)	4 (1.0)
Other	2 (0.7)	4 (1.3)	4 (1.3)	1 (0.4)	2 (0.8)	1 (0.4)	4 (1.0)	5 (1.3)	5 (1.3)

**Table 7 nutrients-14-00848-t007:** Actions taken when barriers to including a vegetable listed on a convenience cooking product back-of-pack recipe arises, by number of categories of convenience cooking product used.

	Use One Product Class			Use Two Product Classes			Use Three Product Classes		
	Do Not Have	Do Not Like	Cannot Consume	Do Not Have	Do Not Like	Cannot Consume	Do Not Have	Do Not Like	Cannot Consume
	n (%)								
Not include that vegetable	18 (9.0)	15 (7.6)	20 (10.1)	18 (9.5)	16 (8.5)	23 (12.2)	6 (5.7)	11 (10.4)	10 (9.4)
Replace (different vegetable)	105 (53.0)	124 (62.6)	113 (57.1)	81 (42.9)	105 (55.6)	98 (51.9)	53 (50.0)	69 (65.1)	69 (65.1)
Replace (similar vegetable)	68 (34.3)	51 (25.8)	51 (25.8)	81 (42.9)	58 (30.7)	53 (28.0)	43 (40.6)	22 (20.1)	24 (22.6)
Not make the meal	4 (2.0)	5 (2.5)	9 (4.5)	6 (3.2)	5 (2.6)	8 (4.2)	3 (2.8)	4 (3.8)	3 (2.8)
Not sure	0 (0.0)	0.0 (0.0)	1 (0.5)	1 (0.5)	1 (0.5)	4 (2.1)	1 (0.9)	0 (0.0)	0 (0.0)
Other	3 (1.5)	3 (1.5)	4 (2.0)	2 (1.1)	4 (2.1)	3 (1.5)	0 (0.0)	0 (0.0)	0 (0.0)

**Table 8 nutrients-14-00848-t008:** Factors influencing vegetable choices by use of key convenience cooking products.

	Meal and Recipe Bases					Marinades					Cooking Sauces				
	Users		Non-Users			Users		Non-Users			Users		Non-Users		
Variable	Mean (95%CI)	Rank	Mean (95%CI)	Rank	*p*	Mean (95%CI)	Rank	Mean (95%CI)	Rank	*p*	Mean (95%CI)	Rank	Mean (95%CI)	Rank	*p*
Flavour	3.19 (3.21–3.38)	1	3.17 (3.11–3.24)	2	0.9	3.25 (3.15–3.35)	1	3.20 (3.14–3.26)	2	0.4	3.18 (3.11–3.25)	2	3.19 (3.12–3.27)	2	0.2
Taste	3.17 (3.1–3.38)	2	3.19 (3.12–3.25)	1	0.8	3.22 (3.11–3.33)	2	3.21 (3.16–3.27)	1	0.9	3.24 (3.17–3.31)	1	3.25 (3.18–3.32)	1	0.9
Quality	3.00 (2.81–3.19)	3	3.00 (2.94–3.07)	3	0.9	3.07 (2.97–3.17)	3	3.01 (2.95–3.08)	3	0.3	3.03 (2.95–3.11)	3	3.03 (2.96–3.10)	3	0.6
Availability	2.9 (2.86–3.05)	4	2.90 (2.83–2.97)	4	0.9	2.96 (2.85–3.06)	4	2.90 (2.83–2.97)	4	0.4	2.93 (2.85–3.00)	4	2.90 (2.82–2.98)	4	0.8
Nutritional Content	2.76 (2.61–2.84)	5	2.76 (2.68–2.84)	5	1.0	2.79 (2.66–2.92)	5	2.74 (2.66–2.81)	5	0.5	2.76 (2.67–2.86)	5	2.74 (2.66–2.83)	5	0.3
Value for Money	2.38 (2.3–2.53)	6	2.36 (2.27–2.44)	7	0.9	2.46 (2.33–2.58)	6	2.38 (2.30–2.47)	6	0.9	2.44 (2.35–2.53)	6	2.35 (2.21–2.45)	6	0.07
Texture	2.36 (2.25–2.51)	7	2.38 (2.29–2.47)	6	0.9	2.39 (2.23–2.52)	7	2.34 (2.26–2.34)	7	0.1	2.42 (2.31–2.53)	7	2.32 (2.22–2.41)	7	0.3
Easy to cook	2.14 (2.16–2.43)	8	2.14 (2.05–2.23)	8	1.0	2.23 (2.09–2.37)	8	2.17 (2.08–2.26)	8	0.5	2.22 (2.11–2.33)	8	2.15 (2.05–2.26)	8	0.3
Cost	2.03 (1.79–2.27	9	2.03 (1.94–2.11)	9	1.0	2.11 (1.98–2.25)	9	2.04 (1.95–2.21)	9	0.3	2.10 (2.00–2.21)	9	2.11 (2.00–2.21)	9	0.2
Sustainability	2.00 (1.79–2.24)	10	1.97 (1.83–2.19)	11	0.9	2.08 (1.99–2.20)	10	2.00 (1.91–2.08)	10	0.3	2.06 (1.97–2.06)	11	1.97 (1.88–2.07)	10 *	0.2
Shelf-life	1.97 (1.83–2.19)	11	2.00 (1.91–2.08)	10	0.9	2.08 (1.94–2.21)	11	1.99 (1.91–2.07)	11	0.2	2.07 (1.96–2.17)	10	1.97 (1.07–1.87)	10 *	0.02
Colour	1.80 (1.72–1.88)	12	1.80 (1.70–1.89)	12	1.0	1.78 (1.68–1.87)	12	1.79 (1.64–1.79)	12	0.9	1.87 (1.76–1.99)	12	1.70 (1.60–1.80)	12	0.9

* marks tied ranks.

**Table 9 nutrients-14-00848-t009:** Factors influencing vegetable choices by number of convenience cooking product categories used.

	Use No Product Class		Use One Product Class		Use Two Product Classes		Use Three Product Classes		
Variable	Mean (95%CI)	Rank	Mean (95%CI)	Rank	Mean (95%CI)	Rank	Mean (95%CI)	Rank	*p*
Flavour	3.16 (3.08–3.24)	2	3.20 (3.10–3.31)	1	3.26 (3.15–3.37)	2	3.30 (3.17–3.44)	1	0.4
Taste	3.19 (3.11–3.27)	1	3.18 (3.07–3.28)	2	3.29 (3.18–3.40)	1	3.25 (3.11–3.40)	2	0.3
Quality	3.01 (2.92–3.09)	3	3.02 (2.90–3.13)	3	3.05 (2.94–3.15)	3	3.10 (2.94–3.27)	3	0.7
Availability	2.89 (2.80–2.98)	4	2.90 (2.78–3.01)	4	2.94 (2.82–3.05)	4	2.98 (2.83–3.13)	4	0.8
Nutritional Content	2.72 (2.63–2.82)	5	2.81 (2.69–2.94)	5	2.72 (2.58–2.87)	5	2.76 (2.58–2.95)	5	0.7
Value for Money	2.31 (2.20–2.42)	7	2.38 (2.24–2.53)	6	2.41 (2.28–2.54)	7	2.50 (2.33–2.67)	6	0.3
Texture	2.36 (2.25–2.48)	6	2.33 (2.17–2.49)	7	2.51 (2.36–2.67)	6	2.30 (2.10–2.51)	7	0.3
Easy to cook	2.11 (2.00–2.23)	8	2.19 (2.03–2.35)	8	2.28 (2.11–2.44)	8	2.25 (2.04–2.45)	8	0.4
Cost	1.99 (1.88–2.11)	9	2.11 (1.96–2.25)	9	2.06 (1.92–2.21)	10 *	2.18 (1.96–2.40)	9	0.4
Sustainability	1.95 (1.83–2.06)	11	1.94 (1.79–2.10)	11	2.19 (2.05–2.33)	9	2.04 (1.86–2.22)	11	0.04
Shelf-life	1.97 (1.86–2.08)	10	2.01 (1.86–2.15)	10	2.06 (1.90–2.21)	10 *	2.12 (1.93–2.32)	10	0.6
Colour	1.71 (1.59–1.83)	12	1.83 (1.67–2.00)	12	1.91 (1.74–2.08)	12	1.69 (1.46–1.91)	12	0.2

* marks tied ranks.

## Data Availability

Data will be made available upon reasonable request, provided appropriate ethics clearances are obtained.

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
