# Peer review of "Correlations between Convenience Cooking Product Use and Vegetable Intake"

_nutrients, 2022, doi:10.3390/nu14040848_

Round 1

Reviewer 1 Report

The manuscript is reasonably structured. It is written clearly and concisely. The topic is relevant and also interesting for readers on other continents.
Some adjustments and verifications are still needed, and are listed below.

  • line 19: word "with" may be missing between associated & lower.
  • line 50: correct type of the bracket for ref. 8
  • lines 141-142: it is not really clear that the table and data refers only to users of convenience cooking products. Please rewrite to clarify.
  • line 157: Table 3, I suggest to explain in the foot of the table what CI is for.
  • line 180: please check the reported intervals of % for "did not have" and "did not use". They are different than a sum of the two values (replace - different; replace - similar) in the Table 5 suggest.
  • line 187: Please check the number of the table. Should be Table 6 instead?
  • line 194: please explain in the foot of the table what * (asterix) stands for.
  • line 246: please check wording in the first half of the line.

Author Response

The manuscript is reasonably structured. It is written clearly and concisely. The topic is relevant and also interesting for readers on other continents.
Some adjustments and verifications are still needed, and are listed below.

Response: thank you for the review and your attention to detail, apologies for the errors, thank you for allowing us to amend them.

  • line 19: word "with" may be missing between associated & lower.

Response: Thank you for picking up this typographical error. This sentence has now been edited to include additional data requested by another reviewer.  

  • line 50: correct type of the bracket for ref. 8

Response: Thank you, reference has been updated accordingly.

  • lines 141-142: it is not really clear that the table and data refers only to users of convenience cooking products. Please rewrite to clarify.

Response: We have added “in users of convenience cooking products.” to the table heading for clarity.

  • line 157: Table 3, I suggest to explain in the foot of the table what CI is for.

Response: Good point, thank you for this attention to detail, we have added this as a foot note.

  • line 180: please check the reported intervals of % for "did not have" and "did not use". They are different than a sum of the two values (replace - different; replace - similar) in the Table 5 suggest.

Response: Thank you, this error has been fixed.

  • line 187: Please check the number of the table. Should be Table 6 instead?

Response: Thank you, this has been updated (noting table numbers have now changed due to additional data being added).

  • line 194: please explain in the foot of the table what * (asterix) stands for.

Response: The * marks tied ranks. We have added this explanation to the table footer.

  • line 246: please check wording in the first half of the line.

Response: We have edited this section to for clarity. It now reads “While convenience cooking products have the potential to help increase vegetable consumption in users, additional research is needed to provide a deeper insight into consumers attitudes and actions towards vegetable consumption when using these products. A better understanding of usage habits and vegetable consumption drivers may assist industry with adapting these products accordingly to ensure an adequate vegetable intake and variety of vegetable intake.”

Reviewer 2 Report

Overall, this manuscript is clearly written, and results are thoroughly presented and discussed.  I think the primary implication is relevant, and, as stated by the authors, is related to the potential for increasing vegetable intake by reformulating back of pack recipes and/or suggesting side dishes to go along with the main dish.  However, there are some areas that need clarification for interpreting the results.

Methods and Materials

More information is needed about the survey:

  • Who developed the questions – the authors? or Qualtrics with consultation by the authors?  Were they pilot-tested prior to administration?
  • It would be helpful to include more detail about the questions in the survey, particularly those for which outcome data is presented. What were the response options (as was provided in lines 104-107 for factors determining vegetable selection) and which questions were open-ended? How were open-ended responses categorized?

Lines 96-103: Were pasta sauces and any other vegetable-based sauces included as a vegetable? Some people may consider pasta sauce as a vegetable. Were respondents prompted to consider vegetable intake in mixed dishes, including what they might add to a meal/recipe base?

Results

Presumably, some individuals used more than one of the convenience products. So, if they used all three, was their data then presented in each category (e.g. marinade and cooking sauce, and/or meal base) in the tables? This seems potentially confounding.  Additionally, did authors consider grouping individuals by number of products used? It seems that a user of a marinade only might be somewhat different than a person who uses all three. Were there frequent users vs occasional users or could that not be ascertained by the questions asked?  It seems that results might possibly look different if frequency and number of convenience products used were considered in data analysis rather than "users" and "non-users". If data is available, I would strongly suggest including an analysis by number of products used and frequency. If not, I think this should be addressed in the discussion.

Discussion

Only one limitation is addressed in the discussion related to the nature of the sample. I think there are additional limitations to consider related to online survey data collection, assessment of vegetable intake with the question(s) used, how individuals were classified for data analysis, as well as possible limitations related to lack of information that might be relevant as discussed above (e.g. frequency of use of convenience products).  

And what are the strengths of the study?

Author Response

Overall, this manuscript is clearly written, and results are thoroughly presented and discussed.  I think the primary implication is relevant, and, as stated by the authors, is related to the potential for increasing vegetable intake by reformulating back of pack recipes and/or suggesting side dishes to go along with the main dish.  However, there are some areas that need clarification for interpreting the results.

Response: thank you for the review and your thoughtful comments, we hope the amendments are satisfactory. We believe this manuscript is stronger for your observations and truly appreciate this authentic and thorough engagement with the review process.

Methods and Materials

More information is needed about the survey:

  • Who developed the questions – the authors? or Qualtrics with consultation by the authors?  Were they pilot-tested prior to administration?

Response: The majority of questions were developed by the authors. Questions were piloted internally (eg. for recognition of product category names) and scales (always/sometimes/never and Likert scales) were standard scales recommended in the Qualtrics system. We have added the wording in italics as follows to clarify this “The online questionnaire used both qualitative and quantitative questions developed by the authors and piloted internally.”

The vegetable variety score was based on a list of the most commonly consumed vegetables in Australia used in the Australian Healthy Eating Quiz the following statement and appropriate reference have been added “As an indicator of vegetable variety and consumption, eating frequency of fourteen of the most commonly consumed vegetables in Australia was assessed, based on the Australian Healthy Eating Quiz [27].”

The list of Factors involved in determining vegetable selection were adapted from Barrett et al (to add value for money) to make this clear we have edited this section of the methods to read “Factors involved in determining vegetable selection were rated on a 5-point Likert scale (ranging from 1 = Not at all important to 5 = extremely important). There were 12 factors rated (taste, flavour, costs, sustainability, availability, texture, colour, nutritional content, easy to cook, shelf life, quality [28] and value for money).”

As the open-ended responses (free answer when choosing other) were extremely limited and not reported, we have removed reference to this in the methods.

  • It would be helpful to include more detail about the questions in the survey, particularly those for which outcome data is presented. What were the response options (as was provided in lines 104-107 for factors determining vegetable selection) and which questions were open-ended? How were open-ended responses categorized?

Response: We appreciate this and so have added more detail to the methods to clarify the questions asked as outlined below.

Regarding user/non-user classification we have revise this section to read: “The participants were asked to self-identify if they used convenience cooking products (Do you use any of the following products?) with several product types listed (to ensure recognition across different brands) and collapsed into the following categories for analysis meal and recipe bases (including meal bases, recipe bases and recipe concentrates), ready-made marinades (marinades only) and convenience cooking sauces (including simmer sauces, pasta sauces and other sauces), or none of the above. Participants were then categorized as “users” or “non-users” of these products.

Regarding recipe following habits we believe the statement “Regarding back-of-pack recipe following participants were asked if they always, sometimes or never follow the recipes provided with products.” clearly reflects the question used.

Regarding actions when recipe following and not having/liking/or could eat a vegetable we have revised this section to read “In a matrix style question, users were also asked to select from a list of actions what they were most likely to do if they were following a back-of-pack recipe on the product classes they reported using and either didn’t have, didn’t like or couldn’t eat a vegetable listed (not include that vegetable, replace it with a different vegetable, replace it with a similar vegetable, not make the meal at all, not sure, other).”

Regarding reported regular vegetable intake we have edited the section to read “Participants were asked to report their typical daily vegetable consumption in serves per day (with a serves guide provided) using the question “How many serves of vegetables do you typically eat per day (1 serve of vegetables is half a cup of cooked vegetables or 1 cup of salad)?”) and a numerical dropdown provided.  Responses were accordingly categorized into either meeting or not meeting the daily recommendation for vegetables as per the Australian Guide to Healthy Eating.”

Regarding the vegetable variety score, as stated in the previous response we have added “As an indicator of vegetable variety and consumption, eating frequency of fourteen of the most commonly consumed vegetables in Australia was assessed, based on the Australian Healthy Eating Quiz [27].” To “Participants were then given a vegetable variety score (score 1 for each vegetable they consumed at least one serve of regularly i.e. at least weekly) with a possible range of scores from 0-14.”

Lines 96-103: Were pasta sauces and any other vegetable-based sauces included as a vegetable? Some people may consider pasta sauce as a vegetable. Were respondents prompted to consider vegetable intake in mixed dishes, including what they might add to a meal/recipe base?

Response: There was no prompt as to what to consider as a vegetable. We have added this as a limitation in the discussion in the following statement “Other limitations include the self-reporting of vegetable intake, which is vulnerable to over-reporting [34], without prompts as to what to consider as a vegetable, which may conversely result in under-reporting”

Results

Presumably, some individuals used more than one of the convenience products. So, if they used all three, was their data then presented in each category (e.g. marinade and cooking sauce, and/or meal base) in the tables? This seems potentially confounding.  Additionally, did authors consider grouping individuals by number of products used? It seems that a user of a marinade only might be somewhat different than a person who uses all three. Were there frequent users vs occasional users or could that not be ascertained by the questions asked?  It seems that results might possibly look different if frequency and number of convenience products used were considered in data analysis rather than "users" and "non-users". If data is available, I would strongly suggest including an analysis by number of products used and frequency. If not, I think this should be addressed in the discussion.

Response: Frequency of use for individual product types was not available, and conducting this analysis would require a larger cohort, as segregation the user groups would result in groups too small for analysis. We have added this as a limitation in the discussion using the following statement “Classification as users and non-users based on self-reporting also lacks resolution as frequency of use was not captured, and this may impact results. This requires further investigation.” We are now attempting to investigate this very question in a new sample.

However, we do have the data to compare by number of products used. We have therefore added to the methods “The number of product categories used was also summed.”

And subsequently to the results we have added “Regarding the number of individual convenience cooking product classes used, 41% of participants used none, 23.5% used only one product type, 22.5% used two product types and 12.6% used all three.”

AND regarding vegetable intake and variety “Those who did not report using any of the convenience cooking products had higher vegetable intakes than those who used two or three product classes (Table 5). However, vegetable variety score did not vary by number of product categories used (Table 5). Those who did not report using any of the convenience cooking products were the most likely to meet the daily recommended intakes of vegetables (19.5%) compared to those using one product class (11.1%), two product classes (13.8%), or three product classes (7.6%; χ2 = 12.8, p = 0.005).“

AND regarding decisions when don’t have/don’t like/can’t eat “These results were similar when analysed by number of convenience cooking product categories used (Table 7.)”

AND regarding vegetable decision factors we have edited the paragraph to read “The top three determinants of choice regardless of convenience cooking product use or number of categories of product used were taste, flavour, and quality (Table 8 &9). The three least important determinants were shelf-life, sustainability, and colour (Tables 8 &9). There were no significant differences in the mean levels of importance given to each factor in determining vegetable choices between those who reported using each convenience cooking product and those who did not, other than a small difference in the importance score for shelf-life in convenience cooking sauce users compared to non-users (Table 8) and sustainability between the number of convenience cooking products used (Table 9)”

Subsequently we have edited the following sections in the discussion to refer also to number of product categories used “This study is the first to investigate the relationships between usage habits key examples of convenience cooking products, and vegetable consumption habits. It appears that the use of these products, both in binary terms (user relative to non-users) and in terms of using a higher number of convenience cooking product categories, was associated with lower vegetable intakes.”

Any additional breakdown (specific combinations such as marinades only) would reduce the sample sizes per group too much for robust statistical analysis.

Discussion

Only one limitation is addressed in the discussion related to the nature of the sample. I think there are additional limitations to consider related to online survey data collection, assessment of vegetable intake with the question(s) used, how individuals were classified for data analysis, as well as possible limitations related to lack of information that might be relevant as discussed above (e.g. frequency of use of convenience products).  

Response: Additional limitations have been added “Other limitations include the self-reporting of vegetable intake, which is vulnerable to over-reporting [34], without prompts as to what to consider as a vegetable, which may conversely result in under-reporting. Classification as users and non-users based on self-reporting also lacks resolution as frequency of use was not captured, and this may impact results. This requires further investigation. Frequency of recipe following also requires more resolution in further investigation to quantify potential differences in “sometimes” followers.”

And what are the strengths of the study?

Response: We have added to the discussion” However, strengths include the large sample size, the multiple measures of vegetable consumption (intake and variety), the multiple categories of convenience cooking product investigated, and the novelty of the questions with potential implications for consumers, industry and public health. The findings presented here justify further research into the role convenience cooking products play in diet quality.”

Reviewer 3 Report

The manuscript entitled “Correlations between convenience cooking product use and  vegetable intake” presents interesting issue, however some corrections are needed

  • This is a very interesting topic but some detailed concussion is missing. Authors should emphasize the most prominent findings
  • The abstract should be a single paragraph and should follow the style of structured abstracts, but without headings.
  • A good abstract should include a problem statement, background, methodology, key finding and a conclusion, which assist the reader to understand the study. 
  • ‘77.7% of respondents were female, and the majority had a university level 125 of education (Table 1).’ - A sentence should never start with a number
  • ‘incomes above AU$75,000’ – please provide value in euro or in dollar for international readers
  • Authors should in their discussion include 3 areas: (1) compare gathered data with the results by other authors, (2) formulate implications of the results of their study and studies by other authors, (3) formulate the future areas which should be studied.
  • Authors should present here and discuss the limitations of their study.

Author Response

The manuscript entitled “Correlations between convenience cooking product use and  vegetable intake” presents interesting issue, however some corrections are needed

  • This is a very interesting topic but some detailed concussion is missing. Authors should emphasize the most prominent findings.

Response: To highlight the main findings in the conclusion we have added the following “The presented data, showing lower vegetable intakes with product use, with higher number of product types used, and with more frequent recipe following but little differences in related vegetable choice factors provide an insight into the convenience cooking products user’s vegetable intake, which provides a baseline for future improvements to the product back of pack recipe vegetable content in hopes to see an increase in the user’s vegetable consumption.”

  • The abstract should be a single paragraph and should follow the style of structured abstracts, but without headings.

Response: We have removed the headings

  • A good abstract should include a problem statement, background, methodology, key finding and a conclusion, which assist the reader to understand the study. 

Response: We have rewritten the abstract in light of additional data added following comments from another reviewer and with these principles in mind.

  • ‘77.7% of respondents were female, and the majority had a university level 125 of education (Table 1).’ - A sentence should never start with a number

Response: We have edited this sentence to start with “The sample was mostly female (77.7%)…”

  • ‘incomes above AU$75,000’ – please provide value in euro or in dollar for international readers

Response: We have added “(equivalent to approximately €47000)”

  • Authors should in their discussion include 3 areas: (1) compare gathered data with the results by other authors, (2) formulate implications of the results of their study and studies by other authors, (3) formulate the future areas which should be studied.

Response: While we respect the reviewers preference, there are multiple ways to frame a discussion. We have formulated the discussion as discussion of major findings in context (the novelty of this research question means there is no direct comparisons available), limitations strengths and future directions, implication and overall conclusion.

  • Authors should present here and discuss the limitations of their study.

Response: To the limitations already addressed we have added “Additional limitations have been added “Other limitations include the self-reporting of vegetable intake, which is vulnerable to over-reporting [34], without prompts as to what to consider as a vegetable, which may conversely result in under-reporting. Classification as users and non-users based on self-reporting also lacks resolution as frequency of use was not captured, and this may impact results. This requires further investigation. Frequency of recipe following also requires more resolution in further investigation to quantify potential differences in “sometimes” followers.”